# An Optimized Peptide Antagonist of CXCR4 Limits Survival of BCR–ABL1-Transformed Cells in Philadelphia-Chromosome-Positive B-Cell Acute Lymphoblastic Leukemia

**DOI:** 10.3390/ijms25158306

**Published:** 2024-07-30

**Authors:** Johanna Pohl, Angela Litz, Omar El Ayoubi, Armando Rodríguez-Alfonso, Ludger Ständker, Mirja Harms, Jan Münch, Hassan Jumaa, Moumita Datta

**Affiliations:** 1Institute of Immunology, Ulm University Medical Center, 89081 Ulm, Germanylemir.el-ayoubi@uni-ulm.de (O.E.A.); 2Core Facility Functional Peptidomics, Ulm University Medical Center, 89081 Ulm, Germany; armando.rodriguez-alfonso@uni-ulm.de (A.R.-A.); ludger.staendker@uni-ulm.de (L.S.); 3Core Unit Mass Spectrometry and Proteomics, Ulm University Medical Center, 89081 Ulm, Germany; 4Institute of Molecular Virology, Ulm University Medical Center, 89081 Ulm, Germany; mirja.harms@uni-ulm.de (M.H.); jan.muench@uni-ulm.de (J.M.)

**Keywords:** BCR–ABL1, CXCR4, AMD3100, Imatinib, cell survival, EPI-X4 derivatives

## Abstract

Philadelphia-chromosome-positive acute lymphoblastic leukemia (Ph^+^ ALL) is characterized by reciprocal chromosomal translocation between chromosome 9 and 22, leading to the expression of constitutively active oncogenic BCR–ABL1 fusion protein. CXC chemokine receptor 4 (CXCR4) is essential for the survival of BCR–ABL1-transformed mouse pre-B cells, as the deletion of CXCR4 induces death in these cells. To investigate whether CXCR4 inhibition also effectively blocks BCR–ABL1-transformed cell growth in vitro, in this study, we explored an array of peptide-based inhibitors of CXCR4. The inhibitors were optimized derivatives of EPI-X4, an endogenous peptide antagonist of CXCR4. We observed that among all the candidates, EPI-X4 JM#170 (referred to as JM#170) effectively induced cell death in BCR–ABL1-transformed mouse B cells but had little effect on untransformed wild-type B cells. Importantly, AMD3100, a small molecule inhibitor of CXCR4, did not show this effect. Treatment with JM#170 induced transient JNK phosphorylation in BCR–ABL1-transformed cells, which in turn activated the intrinsic apoptotic pathway by inducing *cJun*, *Bim*, and *Bax* gene expressions. Combinatorial treatment of JM#170 with ABL1 kinase inhibitor Imatinib exerted a stronger killing effect on BCR–ABL1-transformed cells even at a lower dose of Imatinib. Surprisingly, JM#170 actively killed Sup-B15 cells, a BCR–ABL1^+^ human ALL cell line, but had no effect on the BCR–ABL1^−^ 697 cell line. This suggests that the inhibitory effect of JM#170 is specific for BCR–ABL1^+^ ALL. Taken together, JM#170 emerges as a potent novel drug against Ph^+^ ALL.

## 1. Introduction

The expression of the BCR–ABL1 fusion protein is a hallmark of Ph^+^ ALL, which affects around one-third of adult ALL cases and ~3–5% of pediatric ALL cases [1]. The oncogenic fusion of the breakpoint cluster region (BCR) at chromosome 22 and the tyrosine-protein kinase ABL1 at chromosome 9 results from a reciprocal chromosomal translocation t(9;22)(q34;q11), which leads to constitutive tyrosine kinase activation [2,3]. The prognosis of the disease was very poor until the advent of tyrosine kinase inhibitors (TKIs) such as Imatinib mesylate [4,5,6]. The current treatment regimen for Ph^+^ ALL generally involves chemotherapy in combination with TKI, followed by allogeneic hematopoietic stem cell transplantation (allo-HCT) [7]. In some cases, the combination of TKI with chemotherapy can achieve long-term remission without allo-HCT. Although the inclusion of TKIs in frontline therapy has revolutionized the outcome of the disease, the emergence of TKI resistance through acquired mutation(s) in the ABL kinase domain poses a significant threat to the prognosis of the disease. Some of these mutations, e.g., T315I, can now be targeted by the advanced third-generation TKI inhibitor Ponatinib [8]. However, the toxicity associated with some of the advanced therapeutics as well as their limited access in many of the clinics worldwide make it challenging for clinicians, especially those in resource-constrained situations [7]. This therefore necessitates the continuous search for new, improved inhibitors as single or combinatorial therapeutics [9].

Previously, our laboratory showed that the G-protein coupled chemokine receptor CXCR4 promotes the survival of BCR–ABL1-transformed mouse B cells [10]. Deletion of CXCR4 results in rapid cell death and the complete absence of colony formation in vitro by these cells. Mechanistically, CXCR4 associates with the interleukin 7 receptor (IL7R) on the surface of BCR–ABL1 cells. This leads to the recruitment of IL7R-associated proteins such as Janus kinase 3 (JAK3) in close proximity to CXCR4, thereby activating the JAK-STAT pathway and stimulating cell survival. The hyper-phosphorylation of JAK1-3 is observed in BCR–ABL1-transformed cells, which is reduced by the inducible deletion of CXCR4 in these cells. Thus, the association between these two receptors is a prerequirement for BCR–ABL- induced cell transformation [10]. 

The above observation raises the question as to whether, similar to deletion, the inhibition of CXCR4 also exerts an inhibitory effect on BCR–ABL1-transformed cells. Several small-molecule and peptide inhibitors are available for CXCR4, of which only AMD3100 (Plerixafor) and BL-8040 are FDA-approved drugs, used in stem cell transplantation [11,12]. Owing to the unfavorable side effects of some of these drugs, there is an urgent need to develop novel CXCR4 antagonists. The endogenous peptide inhibitor of CXCR4 (EPI-X4), a 16 mer peptide derived from human serum albumin, exhibits specific binding to CXCR4 and the subsequent blocking of CXCL12-mediated CXCR4 activation [13]. To improve its efficacy and plasma stability, several derivatives of EPI-X4 have been designed using quantitative structure–activity relationship (QSAR) studies [14,15]. For instance, activity-improved derivative EPI-X4 JM#21 has shown therapeutic efficacies in mouse models of inflammatory diseases [14] as well as in oncologic applications [16]. To further improve the stability of optimized EPI-X4 derivatives, they have been conjugated to long-chain fatty acids, leading to increased serum albumin affinity and, thus, to an improved circulation half-life [17]. Previous work from our laboratory have shown that most of these advanced EPI-X4 derivatives potentially block CXCL12-mediated intracellular calcium release in BCR–ABL1-transformed cells in the nanomolar range [14,17]. However, the effect of these advanced derivatives on the survival of BCR–ABL1 cells has not yet been investigated. 

Thus, in the current study, we tested an array of advanced EPI-X4 derivatives for their efficacy in inhibiting BCR–ABL1 cell growth and compared the effect with those of the small-molecule CXCR4 antagonist AMD3100 and the ABL1 TKI Imatinib. We also explored the signaling cascade alterations induced by these inhibitors and tested if the combination of the EPI-X4 derivative with Imatinib exerted a superior cell killing effect. Furthermore, we tested whether the inhibitors would be effective in human ALL cell lines and primary xenograft cells to evaluate their therapeutic potential.

## 2. Results

### 2.1. Optimized EPI-X4 Derivative JM#170 Potentially Blocks Growth of BCR–ABL1-Transformed Mouse B Cells In Vitro

Previously, we have shown that optimized EPI-X4 derivatives such as the 12-mer peptides WSC02 and JM#21 and fatty-acid-conjugated peptides JM#143, JM#169, JM#170, and JM#192 (Table 1) are potent inhibitors of CXCR4 activation as they block CXCL12-induced calcium signaling in BCR–ABL1-transformed mouse bone marrow (BM) B cells [14,17]. To investigate whether this inhibition of CXCR4 signaling exerts any effect on cell survival in vitro, we treated the BCR–ABL1-transformed cells with increasing concentrations (1, 5, and 10 μM) of the above inhibitors as well as AMD3100, a potent small-molecule CXCR4 antagonist, and measured cell growth using real-time live cell imaging in IncuCyte for 96 h. Imatinib, an ABL1 kinase inhibitor known to kill BCR–ABL1 cells at a concentration of 1 μM, was used as the positive control. BCR–ABL1 cells express intrinsic GFP; thus, the enrichment in GFP-positive cells can be used as a marker of cell growth. As depicted in Figure 1A, treatment with the solvent control DMSO led to the accumulation of GFP^+^ cells over time, while treatment with 1 μM Imatinib completely blocked cell growth (Figure 1A, last panel, and Figure 1D). Among the EPI-X4 derivatives, only JM#170 showed a significant reduction in GFP enrichment at 10 μM compared to DMSO (Figure 1A, second panel, and Figure 1B). The remaining inhibitors had no effect on BCR–ABL1 cell growth, except for JM#169, which showed a significant reduction in cell growth only after 72 h (Appendix A). Interestingly, JM#170 blocked cell growth as early as 24 h, while the effect of the small-molecule inhibitor AMD3100 was only distinct at later time points (72 and 96 h, Figure 1A,C). Of note, the inhibitory effect of AMD3100 was also detectable at a lower concentration of 5 μM (Figure 1C). Taken together, the inhibition of CXCR4 by JM#170 and AMD3100 led to the inhibition of BCR–ABL1 cell growth in vitro.

To understand whether this inhibition of cell growth in the presence of JM#170 and AMD3100 was due to cell death induced by the inhibitors, we incubated the cells with 10 μM JM#170 and AMD3100 for different durations and stained them with viability dye. Imatinib (1 μM) served as the positive control. The results showed that JM#170 induced strong and rapid cell death in BCR–ABL1 cells, with ~70% of cells dying within 1 h of treatment (Figure 1E and Appendix A). Cell viability remained low over time, with a prominent increase at 72 h, which could be explained by the loss of stability of peptide-based inhibitors in the serum-containing media. Imatinib, as previously stated, exerted a strong and persistent killing effect on BCR–ABL1-transformed cells starting from 24 h after treatment. 

Surprisingly, AMD3100 did not induce any loss of cell viability within the experimental time period (Figure 1E). Thus, to understand how AMD3100 blocked BCR–ABL1 cell growth as observed before (Figure 1A,C), we labeled the cells with cell trace cell proliferation dye and treated them with 10 μM each of JM#170 and AMD3100. The level of cell proliferation was measured via the dye dilution method. As depicted in Figure 1F,G, AMD3100 significantly reduced the proliferation of BCR–ABL1 cells as compared to DMSO at 48 and 72 h after treatment. This is in line with our previous finding that AMD3100 blocked cell growth only after ~48 h (Figure 1A,C). Interestingly, JM#170 had no effect on proliferation, as cells that survived JM#170 treatment proliferated equally to DMSO-treated ones (Figure 1F). Thus, the mode of action of the two CXCR4 inhibitors are different—while AMD3100 blocks proliferation, JM#170 induces rapid death of BCR–ABL1 cells. 

To address the hypothesis that JM#170 specifically targets CXCR4, we performed an anti-CXCR4 antibody competition assay. BCR–ABL1-transformed cells were treated with increasing concentrations of JM#170, AMD3100 and Imatinib in the presence of a fixed concentration of anti-CXCR4 antibody, followed by flow cytometric analysis to measure the amount of bound antibody. Both JM#170 and AMD3100 significantly inhibited the antibody binding to CXCR4 (Appendix A). However, JM#170 was effective at a much lower concentration (~10 nM) than AMD3100 (~1 μM). Imatinib, on the other hand, had no effect on antibody binding to CXCR4 as it did not interact with CXCR4 (Appendix A). This suggests that JM#170 specifically targets CXCR4 in BCR–ABL1-transformed B cells. 

### 2.2. JM#170 Does Not Affect the Survival of Untransformed B Cells

To address whether the growth inhibition observed in the presence of JM#170 or AMD3100 was specific for BCR–ABL1 cells or caused by the general toxicity of the molecules, we tested their effects on untransformed mouse BM B cells. Previous reports from our laboratory have demonstrated that BM B cells isolated from *Rag2^−/−^ λ5^−/−^ Spl65^−/−^* triple knockout (TKO) mice are blocked at the pro-B-cell stage due to lack of V(D)J recombination [18,19]. These cells can be cultured in the presence of IL7. Similar to WT BM B cells, the transformation of TKO cells with BCR–ABL1 led to enhanced cell growth compared to that of untransformed (empty pMIG vector-transfected (TKO-EV)) cells (Appendix A), indicating that the TKO mutations do not bring about any unexpected cell phenotype. We therefore used these untransformed TKO-EV cells to investigate the effect of the inhibitors. The cells were similarly treated with increasing concentrations of JM#170 and AMD3100, and 1 μM Imatinib, and they were subjected to real-time imaging in IncuCyte for 96 h. As depicted in Figure 2A–C, neither JM#170 nor AMD3100 had any effect on the survival of TKO-EV cells at any concentration tested. Imatinib, as expected, also had no effect on the TKO-EV cells (Figure 2A,D). It could be argued that CXCR4 surface expression may vary between the cell types, with TKO-EV being low in CXCR4 expression, leading to no effect on CXCR4 inhibition. To address this, we measured the cell surface expression of CXCR4 in BCR–ABL1 and TKO-EV cells. The results indicated that the cells expressed comparable amounts of CXCR4 on the surface (Figure 2E). In addition, we tested the inhibitors in normal WT mouse BM-derived B cells. Neither of the inhibitors had any effect on them (Appendix A). Thus, the growth inhibitory effects of JM#170 and AMD3100 are specific to BCR–ABL1-transformed cells and not due to the general toxicity of the molecules. 

### 2.3. JM#170 Blocks CXCL12-Induced ERK1/2 and PI3K Signaling in BCR–ABL1 Cells

From our previous study, we knew that both JM#170 and AMD3100 block the intracellular calcium response upon CXCR4 activation [14,17]. To check if they also block CXCR4-downstream PI3K and MAP kinase (MAPK) activation, we transiently treated the cells with inhibitors and performed Western blotting to detect AKT and ERK1/2 phosphorylation. In the absence of CXCL12 stimulation, the basal level of S473 phosphorylation of AKT remained low, with no changes observed after the addition of the inhibitors (Figure 3A, left panel; Figure 3B; Appendix A). Similarly, basal ERK1/2 phosphorylation at T202/Y204 also remained low in the absence of CXCL12, although a slight increase was observed in DMSO-, JM#170-, and AMD3100-treated cells compared to untreated cells (Figure 3A, left panel; Figure 3C; Appendix A). This might be expected as DMSO is known to alter cellular homeostasis. However, when stimulated with CXCL12, a strong and rapid increase in AKT and ERK1/2 phosphorylation was observed in DMSO treated cells within 5 min of stimulation, while for both JM#170- and AMD3100-treated cells, this elevation was blocked (Figure 3A, right panel; Figure 3B,C; Appendix A). The phosphorylation level for both AKT and ERK decreased over time; however, they remained high in the control compared to that in JM#170- and AMD3100-treated cells. Thirty minutes after stimulation, both signaling pathways seemed to be downregulated. Interestingly, Imatinib did not affect CXCL12-mediated AKT and ERK activation in the initiation phase (5 min), but the signal terminated rapidly especially for ERK (Figure 3A, right panel; Figure 3B,C; Appendix A). Collectively, these data clearly suggest that JM#170 and AMD3100 inhibit the PI3K and ERK signaling pathways downstream of CXCR4 activation.

### 2.4. JM#170 Induces Phosphorylation of JNK and Triggers Intrinsic Apoptotic Pathway in BCR–ABL1 Cells

While analyzing the activation of CXCR4-downstream MAPK pathways in the presence of the inhibitors, we observed that the treatment with JM#170 transiently upregulated basal JNK phosphorylation in BCR–ABL1 cells compared to that in the control DMSO in the absence of any CXCL12 stimulation (Figure 4A,B, Appendix A). A small but significant increase in JNK phosphorylation was observed 5 min after stimulation, while such upregulation was not observed in AMD3100-treated cells. Notably, Imatinib treatment significantly reduced basal JNK phosphorylation at 5 and 15 min after treatment (Figure 4A,B, Appendix A). JNK is known to be activated under various cellular stress conditions, including treatment with inhibitors, which ultimately leads to cell apoptosis [20]. Therefore, to investigate whether JM#170-mediated JNK activation induced apoptosis in BCR–ABL1 cells, we measured the activation of executioner caspases 3 and 7. As depicted in Figure 4C, a significant increase in caspase 3/7 activation was observed in JM#170-treated cells compared to that in DMSO-treated ones 24 and 48 h after treatment. AMD3100, on the other hand, did not induce any caspase activation. Caspase activation was significantly lower in these cells than in DMSO-treated cells (Figure 4C). Next, we checked caspase 8 activation as a marker of the death-receptor-mediated extrinsic apoptotic pathway. The results suggested that neither JM#170 nor AMD3100 induced caspase 8; rather, they downregulated it in comparison to DMSO (Figure 4D). This indicates that JM#170 triggers the intrinsic apoptosis pathway in BCR–ABL1 cells. This is further corroborated by the observation that the JM#170 treatment led to the significant up-regulation of pro-apoptotic genes downstream of JNK activation (Figure 4E). *Jun* and *Bim* were significantly elevated in the presence of JM#170 24 h after treatment, while *Bax* was mildly upregulated. Interestingly, a significant increase in anti-apoptotic *Bcl2* gene expression was also observed in these cells, which could indicate a compensatory effect to balance the induction of apoptosis. Notably, the expression of the other anti-apoptotic gene *Bcl-xl* was not altered at all (Figure 4E). The inactivation of *FasL* in the JM#170-treated cells again supported the noninvolvement of the death-receptor-mediated extrinsic apoptotic pathway. The overall level of gene expression alteration diminished after 48 h, with still significantly elevated levels of *Bax*, *Bim,* and *Bcl2* in JM#170-treated cells (Figure 4F). AMD3100, on the contrary, did not upregulate any gene expression at any time points (Figure 4E,F). 

Thus, collectively, these data reflect that JM#170 spontaneously activates the JNK pathway and triggers intrinsic apoptosis in BCR–ABL1-transformed cells—a process not elicited by AMD3100. 

### 2.5. Combined Imatinib and JM#170 Treatment Kills BCR–ABL1 Cells More Effectively

Since both JM#170 and Imatinib could effectively kill BCR–ABL1-transformed B cells, we checked whether combination of the two inhibitors had an additive effect. We treated the cells with lower doses of Imatinib (10 and 100 nM instead of 1 μM) in the presence or absence of 10 μM JM#170 and measured the cell growth via real-time live cell imaging in IncuCyte, as stated before. The results suggested that in comparison to the control DMSO, Imatinib alone at 10 nM did not have any effect on BCR–ABL1 cell growth (Figure 5A,B) while at 100 nM, a mild but significant growth inhibition was observed specifically after ~72 h (Figure 5A,C,D, Appendix A). The combination of 10nM Imatinib with 10 μM JM#170 induced a stronger inhibition of cell growth, but the effect was comparable to that of JM#170 alone (Figure 5A,B, dark red and olive green lines). Thus, the inhibition observed here could be attributed to JM#170 alone. However, when co-treated with 100 nM of Imatinib, a complete block of cell growth was observed, which was stronger than that achieved with JM#170 alone (Figure 5A,C,D, Appendix A). Therefore, in combination with 10 μM JM#170, a 10-fold lower dose of Imatinib can completely block BCR–ABL1-transformed cell growth.

### 2.6. JM#170 Is Lethal for BCR–ABL1^+^ Human ALL Cell Line

So far, we tested the activity of CXCR4 inhibitors in transformed mouse B cells. To investigate whether JM#170 is also effective in killing human ALL cell lines, we first screened several ALL cell lines for surface CXCR4 expression. Compared to healthy human-blood-derived B cells, CXCR4 expression was found to be the highest in 697 cells and the lowest in SD1 cells (Appendix A). SupB15 and Tom1 expressed intermediate levels of CXCR4. Therefore, we selected SupB15 cells as the BCR–ABL1^+^ ALL cell line and 697 as the BCR–ABL1^−^ cell line for further analysis. When tested for CXCL12-induced calcium mobilization; both JM#170 and AMD3100 completely blocked the calcium flux in SupB15 (Figure 6A,C) and 697 (Figure 6B,D) cell lines at 1 μM. To further assess the CXCR4-specific effect of the inhibitors, we performed an anti-CXCR4 antibody competition assay, as described before. SupB15 cells were treated with increasing concentrations of JM#170, AMD3100, and Imatinib in the presence of a fixed concentration of anti-CXCR4 antibody 12G5. The result suggested that both JM#170 and AMD3100 significantly inhibited 12G5 binding to CXCR4 (Appendix A), with JM#170 being more potent than AMD3100. Imatinib, as expected, had no effect on 12G5 binding to CXCR4 (Appendix A). This clearly suggests that, similar to BCR–ABL1-transformed cells, JM#170 specifically targets CXCR4 in the human ALL cell line SupB15.

Next, we tested if treatment with JM#170 and AMD3100 at 10 μM and Imatinib at 1 μM concentrations, as used previously, had any effect on the viability of these ALL cell lines over time. As depicted in Figure 6E and Appendix A, in comparison to DMSO-treated cells, JM#170 significantly reduced SupB15 cell survival at 48 and 96 h after treatment. Imatinib exerted a stronger and more persistent killing effect over time, while AMD3100, as also observed for mouse cells, had no effect on SupB15 cell survival (Figure 6E). The 697 cells, on the other hand, were resistant to Imatinib treatment, most likely due to the lack of the BCR–ABL1 fusion protein (Figure 6F). Surprisingly, JM#170 also had no effect on the survival of these cells (Figure 6F). This indicates that the cytotoxic effect of JM#170 is dependent on the presence of the BCR–ABL1 protein. A similar observation was obtained from the growth monitoring of these cells using IncuCyte. We observed that 10 μM JM#170 and 1 μM Imatinib significantly reduced SupB15 cell growth when compared to the DMSO-treated cells (Appendix A, upper panel) but had no effect on the 697 cells (Appendix A lower panel). Notably, the inhibitory effect of JM#170, as observed in SupB15 cells, was lower than that observed in BCR–ABL1-transformed mouse primary pre-B cells. 

In addition to the ALL cell lines, we also assessed the efficacy of JM#170 on primary Ph^+^ ALL cells. To this end, we used ALL blast cells isolated from the spleen of immunodeficient mice that were previously xenografted with Imatinib-resistant Ph^+^ ALL primary cells. The xenograft cells (hCD19^+^ mCD45^−^) were cultured and treated with control DMSO, 10 μM JM#170, and 1 μM Imatinib; cellular growth was measured using real-time imaging using IncuCyte for a period of 6 days. Although the xenograft cells survived in culture, they did not proliferate, as observed from their growth curve in the control (Appendix A, black line). JM#170 clearly exhibited a significant growth inhibitory effect on these cells compared to DMSO, whereas no effect was seen for Imatinib treatment, as the primary samples were resistant to Imatinib (Appendix A). 

Taken together, our data suggest that in addition to BCR–ABL1-transformed mouse B cells, JM#170 is effective in killing the BCR-ABL^+^ human ALL cell line and primary Ph^+^ ALL xenograft cells.

## 3. Discussion

The seven-transmembrane G-protein-coupled chemokine receptor CXCR4 plays an essential role in the homing and retention of tumor cells in the protective bone marrow niches, thereby preventing their culmination by therapeutic agents and increasing the risk of disease relapses [21]. The elevated expression of CXCR4 in hematological malignancies as well as in solid tumors is associated with poor prognosis. Consequently, targeting CXCR4 activation through antagonists leads to the egress of hiding cancer cells from bone marrow to the periphery, a process known as chemosensitization, which therefore possesses great therapeutic potential. Several small molecules, peptides, neutralizing antibodies, as well as antibody–drug conjugates are being tested for their efficacy in antagonizing CXCR4 function in different cancers. For instance, the small-molecule inhibitor AMD3100, the peptide-based inhibitors BL-8040 and LY2510924, and anti-CXCR4 monoclonal antibodies have been used in several clinical trials for patients with acute myeloid leukemia (AML) and ALL, both as monotherapies and in combination with chemotherapy (as reviewed in [22]). Although the trials have demonstrated that targeting the CXCR4 axis in vivo is feasible and safe, the clinical outcome in many cases does not prove to be beneficial compared with that of chemotherapy alone. This therefore warrants new improved CXCR4 antagonists. 

Ph^+^ B-ALL represents a group of ALL cases that expresses the oncogenic BCR–ABL1 fusion protein. Although the invention of Imatinib, a tyrosine kinase blocker of BCR–ABL1, greatly improved the outcome of the disease, resistance to Imatinib is a major problem that can lead to relapse. It is therefore crucial to identify other contributing pathways that can be additionally targeted in this disease. A previous study from our laboratory identified both CXCR4 and IL7R pathways as contributing to the pathology of BCR–ABL1 [10]. The complete deletion of CXCR4 was found to be sufficient to block BCR–ABL1 cell growth in vitro. Therefore, the aim of the current work was to study the effect of CXCR4 inhibition on BCR–ABL1-mediated transformation.

EPI-X4, a 16-amino-acid peptide derived from the proteolysis of serum albumin, is a naturally occurring CXCR4 antagonist that prevents CXCR4-tropic HIV-1 virus entry [13]. Using a molecular docking study to predict the EPI-X4 binding sites to CXCR4 and QSAR to identify modifications that might enhance the binding and functional efficacy of EPI-X4, several optimized EPI-X4 derivatives were generated [14]. Many of these derivatives effectively block the CXCL12-induced calcium mobilization in BCR–ABL1-transformed mouse B cells, showing much higher efficacy than parental EPI-X4, indicating that they could inhibit CXCR4 signaling in these cells [14,17]. 

In the current manuscript, we presented data that the inhibition of CXCR4 by a lipid-conjugated truncated version of EPI-X4, namely JM#170, induced rapid and strong cell death in BCR–ABL1-transformed malignant mouse cells, whereas it remained inactive against the WT B cells. Live cell imaging of BCR–ABL1 cells in the presence of different inhibitors clearly indicated that JM#170 potently blocked cellular growth within 24 h of treatment, while AMD3100 showed an effect at a much later time point (~60 h post treatment, Figure 1A–C). Notably, for cells treated with JM#170, a slight increase in GFP^+^ cell accumulation was observed 72 and 96 h post treatment (Figure 1A). This could be explained by the relatively lower stability of peptide-based inhibitors in serum containing cell culture media. Although the optimized derivatives are significantly more stable in blood plasma than the mother peptide EPI-X4 [17], 72–96 h could still be long. The cells that remained resistant to JM#170 until ~48 h may have started proliferating after the degradation of the inhibitor, which in turn increased the GFP^+^ cell density. Interestingly, neither JM#170 nor AMD3100 had any growth inhibitory effect on untransformed bone-marrow-derived B cells (Figure 2 and Appendix A). This may have arisen from the relatively high rate of cell proliferation observed in the transformed cells compared to the WT cells, which was clearly evident from the growth curve of the two cell types in live cell imaging (Figure 1B black line vs. Figure 2B black line). As the requirement for CXCR4 signaling for survival and proliferation of transformed cells is higher than that of normal cells, the effect of CXCR4 inhibition is also stronger on these cells than on normal cells. This selectivity is a very important aspect for JM#170 to be potentially used as an anticancer peptide drug. 

Although both JM#170 and AMD3100 blocked BCR–ABL1 cell growth (Figure 1A–C), as well as CXCL12-induced AKT and ERK1/2 phosphorylation (Figure 3B,C), the end effect was different for the two inhibitors. While JM#170 effectively killed BCR–ABL1 cells, AMD3100 halted the proliferation, with no cell killing effect observed within the experimental timeline (Figure 1E–G). This clearly indicates that JM#170 has a different mechanism of action other than blocking ligand-induced CXCR4 activation in BCR–ABL1-transformed cells. Indeed, we observed that JM#170 spontaneously induced transient JNK1/2 phosphorylation (Figure 4A,B), leading to the enhanced expression of JNK-downstream genes such as *cJun*, *Bax*, and *Bim* (Figure 4E,F) and triggered the intrinsic apoptotic pathway in BCR–ABL1-expressing cells (Figure 4C,D). JNK, also known as stress-activated protein kinase (SAPK), is activated upon various kinds of cellular stress, including treatment with drugs or signaling pathway inhibitors; depending on the context and specific cell types, it can act as both a pro-survival as well as a pro-apoptotic signaling cascade [20]. In this case, treatment with the inhibitor JM#170 exerted cellular stress, thereby activating the JNK-mediated pro-apoptotic pathway.

The strong cytotoxic effect exerted by JM#170 on BCR–ABL1 cells prompted us to check if it could be used in combination with Imatinib, specifically with a lower dose of Imatinib. This would be particularly interesting in terms of reducing the side effects of Imatinib. Co-treatment of JM#170 at 10 μM with Imatinib at 100 nM (10-fold lower than usual) induced a complete block of BCR–ABL1 cell growth, which was even stronger than that achieved with JM#170 alone (Figure 5A,C,D). Another 10-fold reduction in Imatinib concentration (10 nM) did not impart any effect on BCR–ABL1 cell growth; the co-treatment yielded an effect similar in magnitude to that of JM#170 alone (Figure 5A,B). This suggests that the combinatorial treatment of JM#170 and Imatinib could be very effective against BCR–ABL1-induced transformation, which not only allows the lowering of Imatinib doses but might also be helpful for Imatinib-resistant cases.

In addition to its effect on BCR–ABL1-transformed mouse B cells, JM#170 was also active against human BCR–ABL1^+^ ALL cell line SupB15. Although JM#170 blocked the CXCL12-induced calcium flux in both SupB15 and 697 cell lines (human BCR–ABL1^−^ ALL cell line, Figure 6A–D), the killing effect was only observed in BCR–ABL1^+^ SupB15 cells (Figure 6E, Appendix A). This suggests that the presence of BCR–ABL1 is essential for the inhibitory action of JM#170. It is important to mention that the level of CXCR4 expression on the surface of 697 cells was very high compared to that on other cells, such as SupB15 or healthy human-blood-derived B cells (Appendix A). It is therefore possible that the maximum concentration of JM#170 used here (10 μM), though sufficient to block ligand-induced calcium mobilization, was not enough to impart an ultimate effect on the survival of these cells. Additionally, as pointed out before, the magnitude of the inhibitory effect observed in the SupB15 cell line was lower compared to that of BCR–ABL1-transformed mouse cells. This indicates the robust nature of the ALL cell lines: in addition to BCR–ABL1 transformation, there might be additional growth-promoting and death-evading mechanisms making these cell lines difficult to target. The mouse cells, on the other hand, with a single transformation event, could be highly susceptible, as observed in this study.

Our experiment with the xenografted primary Ph+ ALL cells also indicated an important aspect of the inhibitory effect of JM#170 (Appendix A). The relative growth inhibition of the xenografted cells by JM#170 was much milder compared to its effect on BCR–ABL1-transformed cells or the human ALL cell line. This, in our view, is due to the lack of proliferation ability of these cells under in vitro conditions. As revealed from their growth curve, the xenografted cells could survive in culture but never proliferated (Appendix A). Since CXCR4 is more involved in proliferating cells than in resting ones, the effect of JM#170 may be milder in resting cells, leading to subtle growth inhibition.

In conclusion, we provide evidence that JM#170, a lipid-modified EPI-X4-derived CXCR4 antagonist, is a potent inhibitor of BCR–ABL1-positive ALL cell growth in vitro. An interesting question would be why, among all the derivatives, JM#170 was found to be the most active. Among the derivatives tested in this study, JM#170 has the lowest molecular weight (Table 1). It could be possible that this size is optimum for fitting into the binding pocket of CXCR4, thereby preventing ligand-induced activation. In addition, such tight interaction might trigger non-classical signal pathway activation such as JNK as observed here. Further structural studies might shed light on this finding. As the next step for therapeutic application, the effect of JM#170 should be investigated using in vivo ALL models.

## 4. Materials and Method

### 4.1. Reagents and Optimized EPI-X4 Derivatives

Optimized EPI-X4 peptide derivatives (Table 1) were synthesized as described previously [17]. Briefly, the peptides were synthesized via standard Fmoc solid-phase peptide synthesis using a Liberty Blue microwave synthesizer (CEM Corporation, Matthews, NC, USA) and then purified using reversed-phase high-performance liquid chromatography (Waters, Milford, MA, USA), employing an acetonitrile/water gradient under acidic conditions on a Phenomenex C18 Luna column (particle size 5 µm, pore size 100 Å). Purified peptides were lyophilized on a freeze-dryer (Labconco, Kansas City, MI, USA), and the molecular mass was verified by liquid chromatography–mass spectrometry (LC-MS; Waters, Milford, MA, USA). The peptides were dissolved in dimethyl sulfoxide (DMSO, Sigma-Aldrich, Hamburg, Germany) at a stock concentration of 3 mM and further diluted in phosphate-buffered saline (PBS) /complete cell culture media before usage. AMD3100 octahydrochloride hydrate (#A5602) and Imatinib mesylate (#SML1027) were purchased from Sigma-Aldrich, Hamburg, Germany and dissolved in H_2_O and DMSO respectively to create a 10 mM stock. Human and mouse CXCL12 were purchased from Peprotech, Hamburg, Germany (#300-28A, #250-20A) and dissolved at 100 μg/mL in H_2_O.

### 4.2. Cell Culture

BCR–ABL1-transformed mouse pre-B cells were generated in-house, as described previously [10]. Briefly, wild-type (WT) mouse bone marrow B cells were cultured in Iscove’s modified Dulbecco’s media (Sigma- Aldrich, Hamburg, Germany) supplemented with 10% heat-inactivated fetal bovine serum (FBS, PAN Biotech, Aidenbach, Germany), 2 mM L-glutamine (Gibco, Dreieich, Germany), 100 units/mL of penicillin/streptomycin (Gibco), 50 μM beta-marcaptoethanol (Gibco), and 1 ng/mL of recombinant mouse interleukin 7 (IL7, Immunotools, Friesoythe, Germany) for 4–7 days at 37 °C in a 7.5% CO_2_ incubator. The cells were then retrovirally transduced with a BCR–ABL1 construct cloned in pMIG vector containing an EGFP marker. Two days after transduction, cells were tested for GFP expression via FACS, and positive cells were selected via IL7 withdrawal from culture medium. BCR–ABL1 transduction transforms the cell, leading to uncontrolled growth and eventually enriching the culture. *Rag2^−/−^ λ5^−/−^ Spl65^−/−^* TKO mouse pro-B cells were cultured in similar IL7 containing Iscove’s media at 37 °C in a 7.5% CO_2_ incubator and transduced with either BCR–ABL1 construct (TKO-BCR–ABL1) or empty pMIG vector containing an EGFP marker (TKO-EV). Selection and enrichment of TKO-BCR–ABL1 cells were performed as stated before. Human ALL cell lines SupB15 (ACC 389), Tom1 (ACC 578), SD1 (ACC 366), and 697 (ACC 42) were purchased from DSMZ. SupB15 cells were maintained in McCoy’s 5A (Biowest, Nuaillé, France) and the rest of the cells in RPMI media (Gibco), supplemented with 20% heat-inactivated FBS, 2 mM L-glutamine, and 100 units/mL of penicillin/streptomycin at 37 °C in a 5% CO_2_ incubator. 

### 4.3. Isolation of Ph^+^ ALL Cells Form Xenografted NSG Mice

Xenografting of immunodeficient NOD.Cg-Prkdcscid Il2rgtm1Wjl/SzJ (NSG) mice with an Imatinib-resistant ALL sample from a patient was described in [10]. Animals showing clinical symptoms of leukemia or with >75% leukemic blasts in blood were sacrificed, and total splenic cells were isolated and frozen for future use.

### 4.4. Real-Time Cell Growth Analysis by IncuCyte

A flat-bottom 96-well plate was coated with 0.01% poly-L ornithine (#P4957, Sigma-Aldrich, Hamburg, Germany) for 1 h at room temperature (RT). BCR–ABL1, SupB15, and 697 cells were counted as 0.5 × 10^5^ cells/well, while TKO-BCR–ABL1 and TKO-EV cells as 1 × 10^5^ cells/well, which were labeled with IncuCyte Cytotox Red dye (#4632, Sartorius, Ulm, Germany) as per the manufacturer’s protocol. The cells were plated in 100 μL complete medium. The solvent control DMSO, peptide inhibitors, AMD3100 or Imatinib were diluted in complete medium at 2X concentration and mixed with the cells in a 1:1 ratio. Plate was prewarmed at 37 °C for 30 min prior to placing in the IncuCyte chamber. Scanning was conducted every two hours for total of 4 days. Images were acquired using a 20X objective in phase contrast and fluorescence mode at 4 images/well and analyzed using the cell-by-cell analysis module of IncuCyte software version 2020B (Essen Bioscience, Ann Arbor, MI, USA). 

For Ph^+^ ALL cells isolated from the xenografted NSG mice, frozen cells were thawed and cultured in MEM-alpha (Gibco) media supplemented with 20% FBS and 100 units/mL of penicillin/streptomycin for 5–16 h in a 5% CO_2_ incubator. Next, the cells were overlaid on 10 mL Ficoll-Paque™PLUS (#17144003, Cytiva, Freiburg, Germany) in 50 mL tubes and centrifuged at 400 g without break for 10 min at RT. The cells were collected from the interface of Ficoll and media and washed once with PBS. Viable cells were counted and plated into previously coated 48-well plates at 4.25 × 10^5^ cells/well in 250 μL of media with or without the inhibitors. Scanning was conducted every two hours for total of 6 days in phase contrast mode and analyzed using the cell-by-cell analysis module of IncuCyte software version 2020B (Essen Bioscience). 

### 4.5. Flow Cytometry

Cells treated with either solvent control DMSO or different concentrations of inhibitors for different time points were collected and washed with ice-cold PBS. Cells were then stained with efluor 450-labeled fixable viability dye (FVD, #65-0863-14, eBioscience, San Diego, CA, USA) at 1:1000 dilution at 4 °C or using 50 μg/mL propidium iodide (PI, #P3566, Thermo Fisher, Dreieich, Germany) solution at RT for 15 min in the dark. After washing, labeled cells were resuspended in 100 μL of PBS and acquired using LSR-Fortessa (BD Bioscience, Heidelberg, Germany). FACS data were analyzed with FlowJo software version 10.0.

To determine the expression of CXCR4 on the cell surfaces, 0.5 × 10^6^ BCR–ABL1 and TKO-EV cells were collected, washed with FACS buffer (PBS containing 3% FBS), and stained with a 1:100 dilution of BV421-labeled anti-mouse CXCR4 antibody (#146511, Biolegend, San Deago, CA, USA). Human ALL cell lines were similarly collected and stained with a 1:100 dilution of Alexa flour 488-labeled antihuman CD19 (#53-0199-42, Invitrogen, Dreieich, Germany) and APC-Cy7-labeled anti-human CXCR4 (#306527, Biolegend) antibodies at 4 °C for 15 min in the dark. Cells were then washed with FACS buffer and analyzed using LSR-Fortessa (BD Bioscience).

Human peripheral blood mononuclear cells (PBMCs) were isolated using Ficoll-Paque™PLUS density gradient centrifugation as per the manufacturer’s instructions. Mononuclear cells were collected from the interface of plasma and Ficoll–Paque media, then washed several times with PBS. Cells were then counted and either used directly for experimentation or frozen in 10% DMSO-containing media for future use. 

### 4.6. Cell Proliferation Assay

BCR–ABL1 cells were collected at 0.5 × 10^5^ cells/well of a 96-well plate and labeled with cell trace far red cell proliferation dye (#C34564, Invitrogen, Dreieich, Germany), according to the manufacturer’s instructions. Labeled cells were plated in 100 μL of complete media and were mixed with equal volume of 2X DMSO or inhibitor-containing media. A small part of the labeled cells was measured in FACS to determine the fluorescence signal on day 0. After specific time periods (48 and 72 h), cells were collected, washed with FACS buffer, and analyzed using LSR-Fortessa (BD Bioscience).

### 4.7. Anti-CXCR4 Antibody Competition Assay

An anti-CXCR4 competition assay was conducted as previously described [14,17]. Briefly, 1 × 10^5^ BCR–ABL1-transformed mouse B cells were incubated with serially diluted (10 μM to 1 nM) JM#170, AMD3100, and Imatinib in the presence of a fixed concentration of BV421-labeled antimouse CXCR4 antibody (clone L276F12, #146511, Biolegend) for 2 h at 4 °C. Afterward, the cells were washed to remove unbound inhibitors and antibody, and measured via FACS for the fluorescence signal of the bound antibody. The fluorescence signal corresponding to only antibody (no inhibitor) was considered to be 100% antibody binding. For SupB15 cells, a similar experiment was carried out using APC-Cy7-labeled anti-human CXCR4 antibody (clone 12G5, #306527, Biolegend). 

### 4.8. Calcium Flux Analysis

Ca^2+^ signaling was analyzed as described previously [14]. Briefly, SupB15 and 697 cells were collected at 1 × 10^6^ cells/treatment and loaded with calcium-sensitive dye Indo-1 AM (#I1223, Invitrogen) and 0.5 mg/mL of pluronic F-127 (#P3000MP, Invitrogen) in their respective media supplemented with 1% FBS at 37 °C for 45 min. Cells were then washed and treated with the inhibitors for 10 min at 37 °C. Baseline signal for calcium was measured for 30 s via flow cytometry, followed by stimulation with 100 ng/mL human CXCL12. The area under the curve (AUC) of each calcium flux plot was determined using FlowJo. The AUC of the water control (solvent for CXCL12) was subtracted from each treatment to obtain the correct estimation of the calcium signal upon CXCL12 stimulation.

### 4.9. Western Blotting

BCR–ABL1 cells were collected as 5 × 10^6^ cells/500 μL complete media and treated either with DMSO or different inhibitors in the absence and presence of 100 ng/mL mouse CXCL12 for 5, 15, and 30 min at 37 °C. Cells were then immediately transferred onto ice to stop the reaction and washed with ice-cold PBS. Cell extract was prepared by resuspending the cell pellet in 100 μL of radio-immunoprecipitation assay (RIPA) buffer supplemented with 1X protease inhibitor cocktail (#78429, Thermo Fisher), 1 mM sodium orthovanadate (Sigma-Aldrich), and 10 mM beta-glycerophosphate (Sigma-Aldrich). For each treatment, 15 μL of cell extract was mixed with 5 μL of 4X Laemmli buffer (reducing), boiled at 95 °C for 10 min, and run on 10% SDS-PAGE. Gels were wet-transferred to PVDF membranes (Millipore, Burlington, MA, USA) and stained with 0.1% ponceau-S solution to check for efficient blotting. The membranes were blocked with 3% BSA solution and incubated with primary antibodies at 4 °C overnight. The next day, primary antibodies were washed, and membranes were probed with horseradish peroxidase (HRP)-conjugated secondary antibodies at RT for 2 h. After washing, the blots were developed using Immobilon ECL Ultra Western HRP Substrate (#WBULS0500, Millipore), and images were acquired with a Fusion SL gel imaging system (Vilber Lourmat, Eberhardzell, Germany). The following primary antibodies from Cell Signaling were used: anti-Phospho-Akt (Ser473, #4060), anti-AKT (#9272), anti-Phospho-p44/42 MAPK (Thr202/Tyr204, #4370), anti-p44/42 MAPK (137F5, #4695), anti-Phospho-SAPK/JNK (Thr183/Tyr185) (81E11, #4668), anti-SAPK/JNK (#9252), and anti-GAPDH (14C10, #2118) antibody. Anti-rabbit IgG, HRP-linked (#7074, Cell Signaling technology, Danvers, MA, USA) was used as secondary antibody.

### 4.10. Caspase 3/7 and Caspase 8 Activation Assay

BCR–ABL1 cells were plated at 0.5 × 10^5^ cells/well of a 96-well plate in 100 μL of complete media and treated with DMSO or various inhibitors. At specific time points (24, 48 and 72 h), activation of caspases was measured with a Caspase-Glo^®^ 3/7 Assay (#G8090) and Caspase-Glo^®^ 8 Assay (#G8200, Promega, Madison, WI, USA) following the manufacturer’s instructions.

### 4.11. RNA Isolation and Real-Time Gene Expression Analysis

BCR–ABL1 cells were plated at 1 × 10^6^ cells/well of a 12-well plate in 1 mL of complete media and treated with DMSO or various inhibitors. After specific time points (24 and 48 h), RNA was isolated from cells using a ReliaPrep Cell RNA miniprep kit (#Z6011, Promega). Fifty nanograms of total RNA was reverse-transcribed using a high-capacity RNA to cDNA kit (#4387406, Thermo Fisher), and gene expression was measured via Taqman assay using Taqman Gene expression master mix (#4369016, Thermo Fisher) following ghd manufacturer’s protocol. The following Taqman probes were used: Jun (Mm07296811_s1), Bax (Mm00432051_m1), Bim (Mm00437796_m1), Bcl2 (Mm00477631_m1), Bcl-xl (00437783_m1), FasL (Mm00438864_m1), and beta actin (04394036_g1).

### 4.12. Statistical Analysis

Statistical analysis was performed using GraphPad Prism 6.0 software. Specific statistical tests are mentioned in the figure legends. Unless otherwise mentioned, all experiments were independently conducted at least three times. 

## Figures and Tables

**Figure 1 ijms-25-08306-f001:**
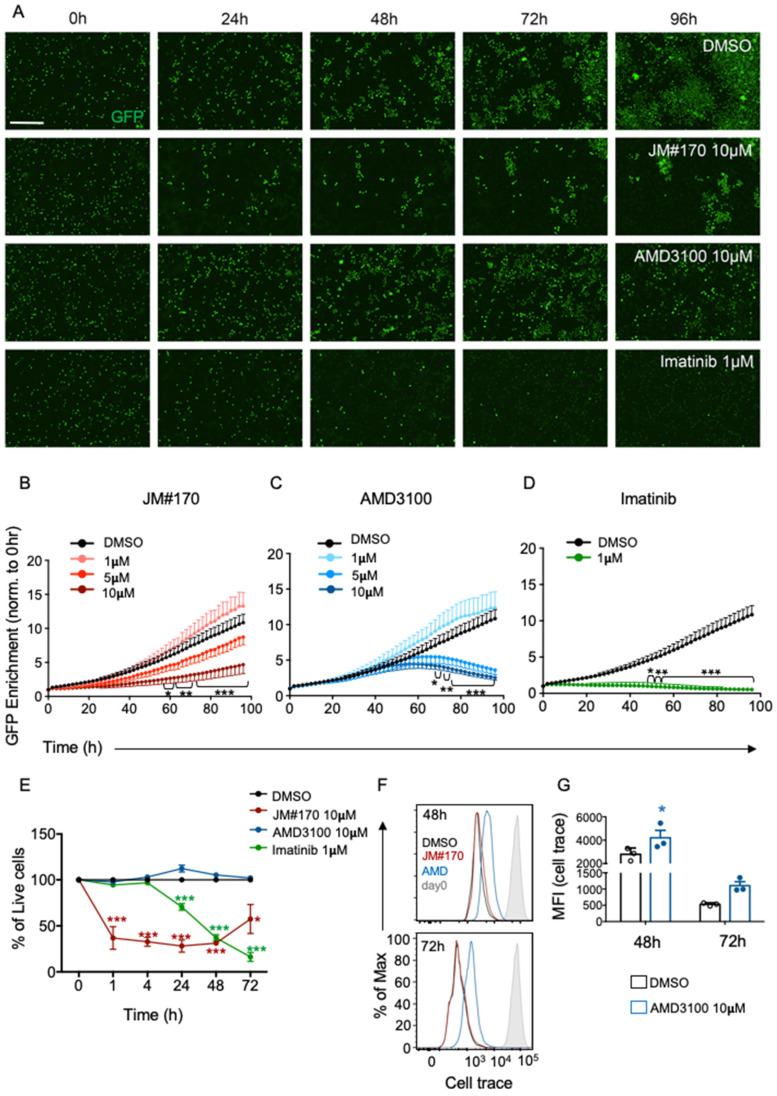
**Effect of JM#170 on cellular growth in BCR–ABL1-transformed mouse B cells.** (**A**) Real-time imaging of BCR–ABL1-transformed mouse B cells treated with solvent (DMSO), 10 µM JM#170 or AMD3100 or 1 µM Imatinib over 96 h. Images are representative of one experiment out of *n* = 4–5. Scale bar: 200µm. (**B**,**C**) Quantification of GFP^+^ cell enrichment as a marker of BCR–ABL1 cell growth in control (DMSO) and different (1, 5, and 10 µM) concentrations of JM#170 (**B**) and AMD3100 (**C**) treated cells over 96 h. The count of GFP^+^ cells for each time point for each treatment was normalized with respect to the corresponding count at 0 h. Graph represents mean ± SEM, *n* = 4–5. (**D**) Similar quantification in control (DMSO) and 1 µM Imatinib-treated cells. Graph represents mean ± SEM, *n* = 5. Statistical analysis—two-way ANOVA with Dunnett’s multiple comparison test. * *p* < 0.05, ** *p* < 0.01, *** *p* < 0.001. (**E**) Flow cytometric analysis of viability of BCR–ABL1 cells treated with solvent (DMSO), 10 µM JM#170 or AMD3100, or 1 µM Imatinib over the indicated time period. The live cell count for each treatment for each time point was normalized with the corresponding live cell count for DMSO and is represented as a percentage. Graph represents mean ± SEM, *n* = 3. Statistical analysis—two-way ANOVA with Sidak’s multiple comparison test. * *p* < 0.05, *** *p* < 0.001. (**F**) Flow cytometric analysis of proliferation of BCR–ABL1 cells treated with DMSO and 10 µM each of JM#170 and AMD3100 for the indicated time period. (**G**) Quantification of cell trace dye dilution, represented by the median fluorescence intensity (MFI) of the dye in DMSO and AMD3100-treated BCR–ABL1 cells after 48 and 72 h of treatment. Bar represents mean ± SEM, *n* = 3. Statistical analysis—two-way ANOVA with Sidak’s multiple comparison test. * *p* < 0.05.

**Figure 2 ijms-25-08306-f002:**
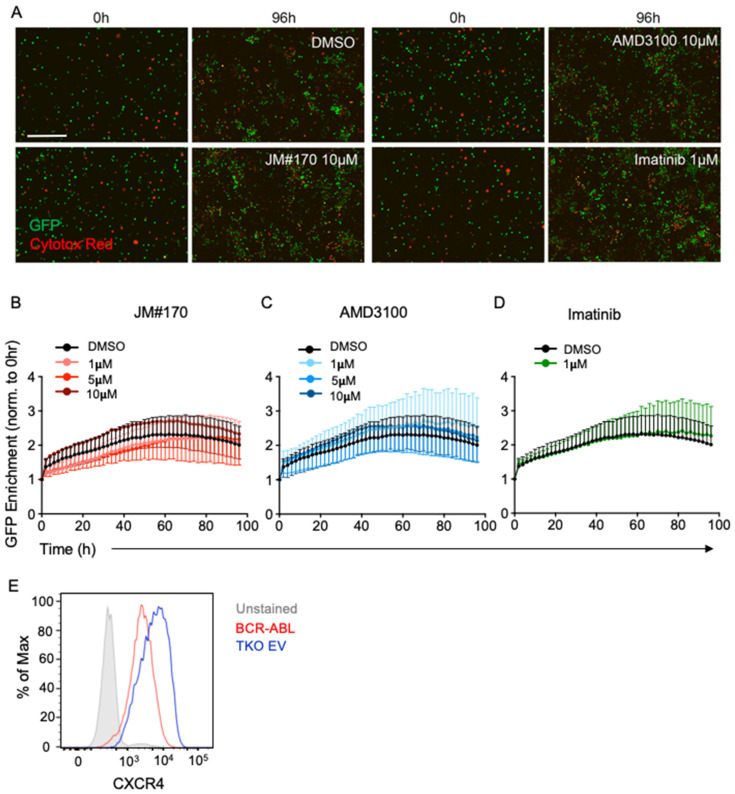
**Effect of JM#170 on cellular growth in untransformed mouse bone marrow B cells.** (**A**) Real-time imaging of bone marrow B cells derived from *Rag2^−/−^ λ5^−/−^ Slp65^−/−^* TKO mice treated with solvent (DMSO), 10 µM of JM#170 or AMD3100, or 1 µM Imatinib over 96 h. Cells expressed intrinsic GFP and were labeled with Cytotox Red dye to measure cell death. Images are representative of one experiment of *n* = 3. Scale bar: 200 µm. (**B**,**C**) Quantification of GFP^+^ cell enrichment as a marker of TKO cell growth in control (DMSO) and treatments with different (1, 5, and 10 µM) concentrations of JM#170 (**B**)- and AMD3100 (**C**) over 96 h. The count of GFP^+^ cells for each time point for each treatment was normalized with respect to the corresponding count at 0 h. Graph represents mean ± SEM, *n* = 3. (**D**) Similar quantification in control (DMSO) and 1 µM Imatinib-treated TKO cells. Graph represents mean ± SEM, *n* = 3. (**E**) Histogram showing CXCR4 expression on the surface of BCR–ABL1 (red line) and TKO (blue line) cells. Grey-filled histogram represents unstained sample. Statistical analysis—two-way ANOVA with Dunnett’s multiple comparison test.

**Figure 3 ijms-25-08306-f003:**
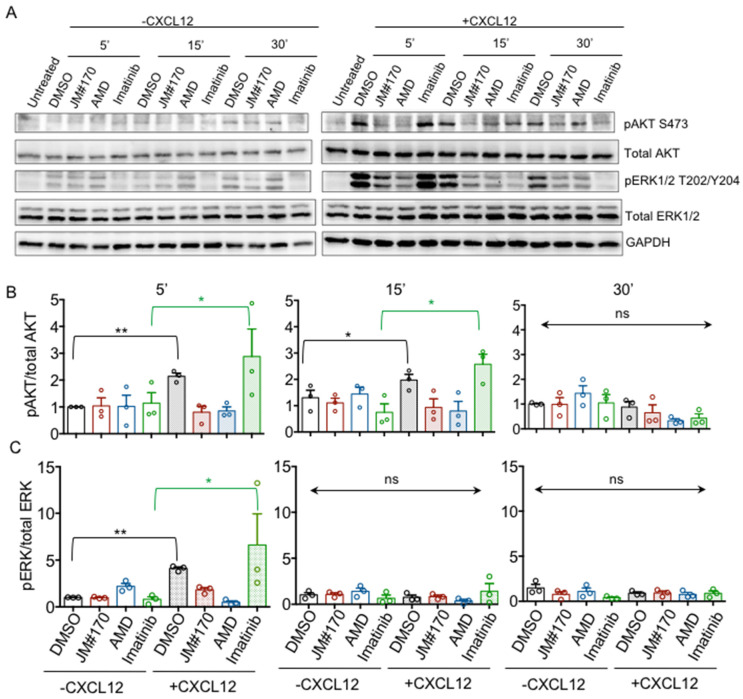
**Inhibitory effect of JM#170 on CXCL12-induced CXCR4-downstream signal activation in BCR–ABL1 cells.** (**A**) Western blot analysis depicting the level of AKT and ERK1/2 phosphorylation in BCR–ABL1 cells treated with DMSO, 10 µM of JM#170 or AMD3100, and 1 µm Imatinib for the indicated time period in the absence (left panel) and presence (right panel) of CXCL12 stimulation. GAPDH was used as the loading control. Image representative of *n* = 3 independent experiments. (**B**,**C**) Quantification of AKT (**B**) and ERK1/2 (**C**) phosphorylation for the Western blotting presented in (**A**). The band intensity of the phosphoprotein was normalized with respect to the total protein for each treatment and is represented as fold change with respect to 5 min DMSO (considering 5 min DMSO as 1). Bar represents mean ± SEM, *n* = 3. Each circle represents one individual experiment. Statistical analysis—one-way ANOVA with Sidak’s multiple comparison test. * *p* < 0.05, ** *p* < 0.01. ns: not significant.

**Figure 4 ijms-25-08306-f004:**
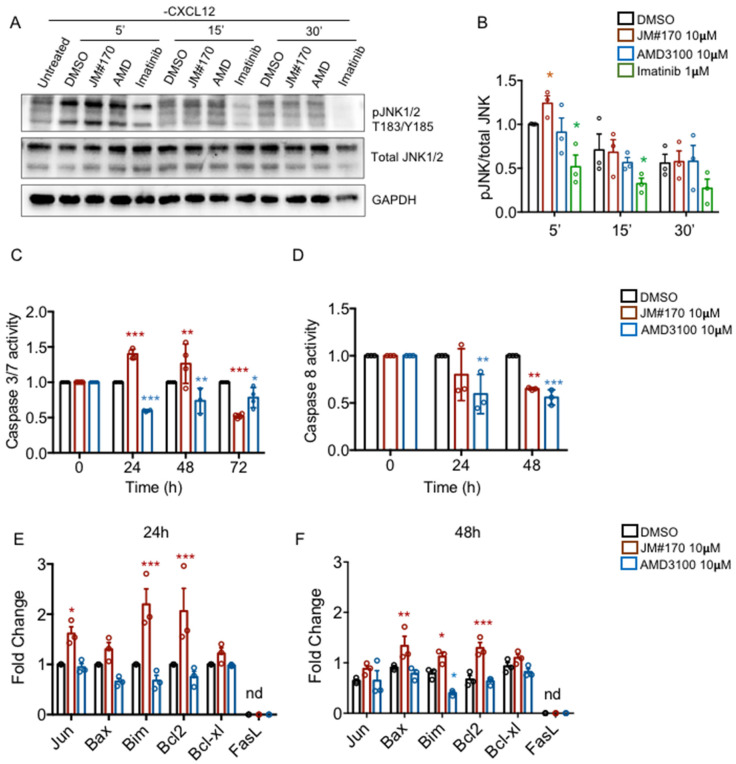
**JM#170 induces intrinsic apoptosis in BCR–ABL1 cells.** (**A**) Western blot analysis depicting the level of JNK1/2 phosphorylation in BCR–ABL1 cells treated with DMSO, 10 µM of JM#170 or AMD3100, and 1 µM Imatinib for the indicated time period in the absence of any CXCL12 stimulation. GAPDH was used as the loading control. Image representative of *n* = 3 independent experiments. (**B**) Quantification of JNK1/2 phosphorylation, as shown in (**A**). The band intensity of the phosphoprotein was normalized with respect to the total protein for each treatment and is represented as fold change with respect to 5 min DMSO (considering 5 min DMSO as 1). Bar represents mean ± SEM, *n* = 3. Each circle represents one individual experiment. Statistical analysis—two-way ANOVA with Dunnett’s multiple comparison test. * *p* < 0.05. (**C**,**D**) Analysis of caspase 3/7 (**C**) and caspase 8 (**D**) activity in BCR–ABL1 cells treated with DMSO or 10 µM each of JM#170 and AMD3100 for the indicated time period. Values of each time point were normalized with respect to the corresponding DMSO. Bar represents mean ± SEM, *n* = 3. Statistical analysis—two-way ANOVA with Dunnett’s multiple comparison test. * *p* < 0.05, ** *p* < 0.01, *** *p* < 0.001. (**E**,**F**) Real-time gene expression analysis depicting the relative expression of the pro/anti-apoptotic genes in BCR–ABL1 cells treated with DMSO, 10 µM JM#170 or AMD3100 for 24 (**E**) and 48 h (**F**). Bar represents mean ± SEM, *n* = 3. Statistical analysis—two-way ANOVA with Dunnett’s multiple comparison test. * *p* < 0.05, ** *p* < 0.01, *** *p* < 0.001. nd: not detected.

**Figure 5 ijms-25-08306-f005:**
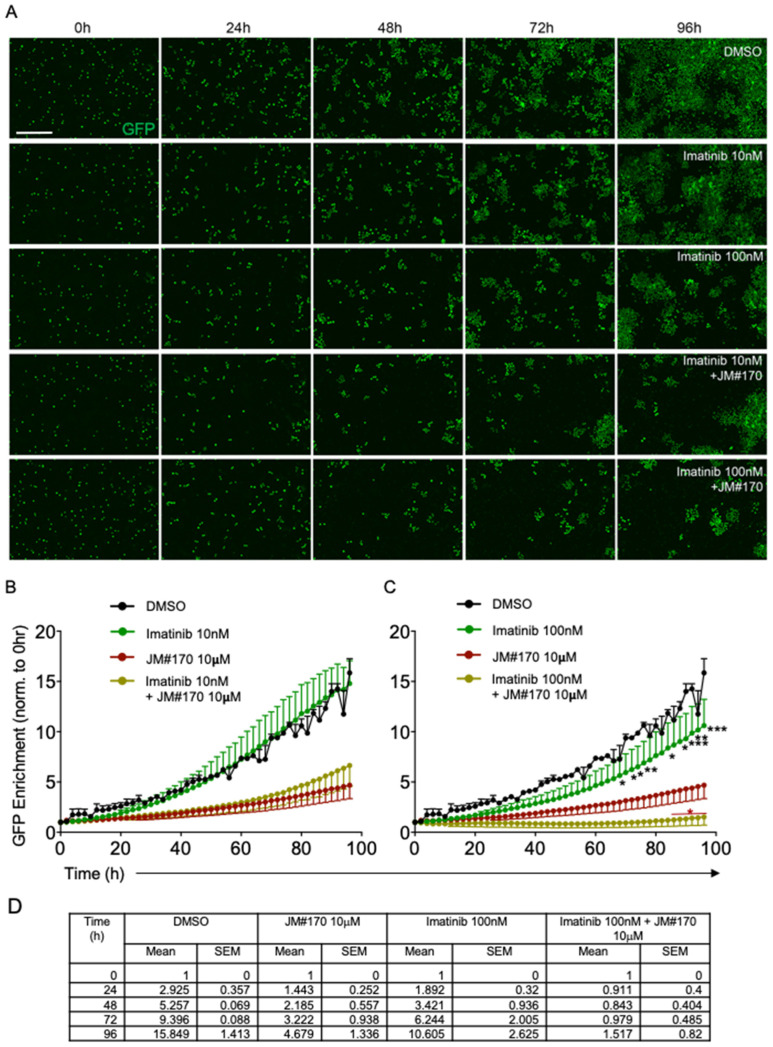
**JM#170 acts synergistically with Imatinib to strongly inhibit BCR–ABL1 cell survival.** (**A**) Real-time imaging of BCR–ABL1 cells treated with DMSO, 10 or 100 nM Imatinib, and 10 or 100 nM Imatinib in combination with 10 µM JM#170 over 96 h. Images are representative of three independent experiments. Scale bar: 200 µm. (**B**) Quantification of GFP enrichment as a marker of BCR–ABL1 cell growth in DMSO, 10 nM Imatinib, and 10 nM Imatinib in combination with 10 µM JM#170 treated cells over 96 h. The count of GFP^+^ cells for each time point for each treatment was normalized with respect to the corresponding count at 0 h. The 10 µM JM#170 plot (dark red) was taken from Figure 1B to facilitate comparison. (**C**) Similar quantification of GFP enrichment in DMSO, 100 nM Imatinib, and 100 nM Imatinib in combination with 10 µM JM#170. Graph represents mean ± SEM, *n* = 3. Statistical analysis—two-way ANOVA with Dunnett’s multiple comparison test. Black *: comparison with respect to DMSO, dark red *: comparison with JM#170 alone. * *p* < 0.05, ** *p* < 0.01, *** *p* < 0.001. (**D**) Table representing the normalized GFP+ cell counts for the different treatment groups at the indicated time points.

**Figure 6 ijms-25-08306-f006:**
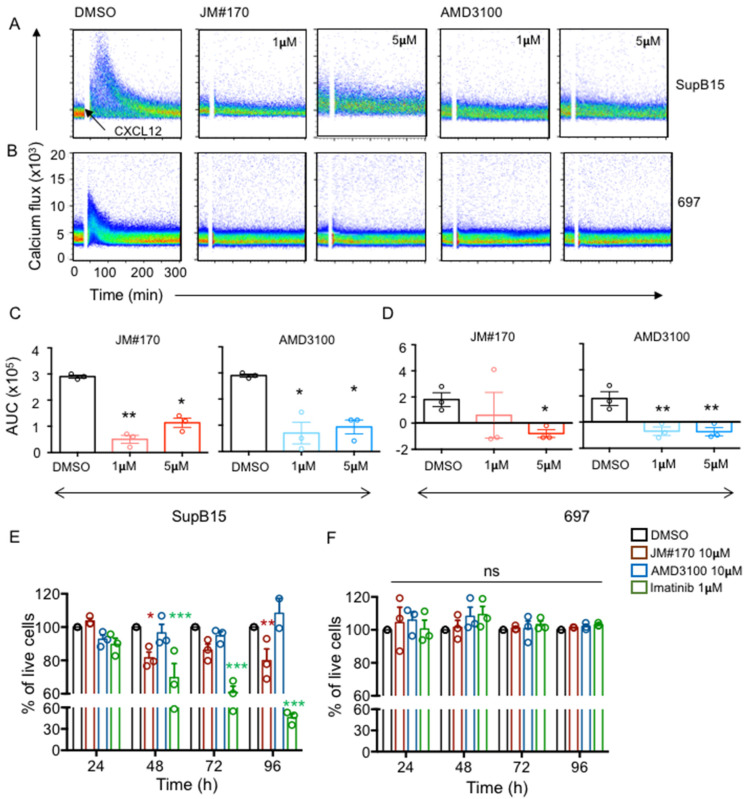
**JM#170 efficiently blocks CXCR4 activation in BCR–ABL1-positive human ALL cell line and induces cell death.** (**A**,**B**) Flow cytometric measurement of CXCL12-driven calcium flux in SupB15 (**A**) and 697 (**B**) cells treated with solvent (DMSO), 1 or 5 µM JM#170, and 1 or 5 µM AMD3100. The baseline was measured for 30 s, which was followed by the addition of 100 ng/mL human CXCL12 (black arrow), and the signal was recorded for a total of 5 min. Images representative of three independent experiments. (**C**,**D**) Quantification of the above calcium flux analysis for SupB15 (**C**) and 697 (**D**) cells treated with DMSO (black), 1 µM (light red) and 10 µM (red) JM#170 and 1 µM (light blue) and 10 µM (blue) AMD3100. The area under the curve (AUC) for each measurement was calculated, and the AUC of the water control (solvent control of CXCL12) was subtracted from each measurement. The subtracted values are plotted as mean ± SEM, *n* = 3. Statistical analysis—one-way ANOVA with Dunnett’s multiple comparison test. * *p* < 0.05, ** *p* < 0.01. (**E**,**F**) Flow cytometric analysis of cell survival of SupB15 (**E**) and 697 (**F**) cells treated with solvent DMSO, 10 µM JM#170 or AMD3100, or 1 µM Imatinib over the indicated time period. The live cell count for each treatment for each time point was normalized to the corresponding live cell count for DMSO and is represented as a percentage. Graph represents mean ± SEM, *n* = 3. Each circle represents one individual experiment. Statistical analysis—two-way ANOVA with Dunnett’s multiple comparison test. * *p* < 0.05, ** *p* < 0.01, *** *p* < 0.001.

**Table 1 ijms-25-08306-t001:** Sequence of the optimized EPI-X4 derivatives.

EPI-X4 Derivative	Sequence	Molecular Weight (Da)	Reference
Original	LVRYTKKVPQVSTPTL	1832	[13]
WSC02	IVRWSKKVPCVS	1401	[13]
JM#21	ILRWSRKLPCVS	1458	[14]
JM#143	ILRWSRK * (Glu-Pal)LPCVS	1825	Harms et al., 2024 [17]
JM#169	IVRWSKK * (Pal)VPCVS	1640	Harms et al., 2024 [17]
JM#170	ILRWSRK * (Pal)-NH_2_	1197	Harms et al., 2024 [17]
JM#192	ILRWSRK * (Glu-Pal)-NH_2_	1325	Harms et al., 2024 [17]

* Modified residue—C16 long-chain fatty acid (palmitic acid) linked to the ε-amino side chain of K either directly or via a glutamic acid linker.

## Data Availability

All data supporting the findings of this study are available within the article and its Appendix A.

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
