# Peer review of "An Optimized Peptide Antagonist of CXCR4 Limits Survival of BCR–ABL1-Transformed Cells in Philadelphia-Chromosome-Positive B-Cell Acute Lymphoblastic Leukemia"

_ijms, 2024, doi:10.3390/ijms25158306_

Round 1

Reviewer 1 Report

Comments and Suggestions for Authors

Hassan and Moumita et al. submitted the manuscript entitled: An optimized peptide antagonist of CXCR4 limits survival of BCR-ABL1 transformed cells in Philadelphia chromosome-positive B cell acute lymphoblastic leukemia, in which the authors developed an oligopeptide, named JM#170, that can target CXCR4 and effectively inhibit the Bcr-Abl fusion protein-driven cancer cells. The authors also tested downstream pathways of CXCR4 and observed protein expression regulation of several CXCR4-related proteins. While this topic will be of interest to potential readers of IJMS, I believe the manuscript need extensive edit to better support the authors’ findings and conclusions.

My comments are as follows.

1. Table 1, it is suggested to label * on the exact residue with modification.

2. Figure 1: All the figures in figure 1 did show cytotoxicity of JM#170 but not related to CXCR4 inhibition.

3. Figure 1c: From the current version, AMD3100 showed inhibitory ability at 5 and 10 uM, which contradict with figure 1e.

4. Page 5, line 154-156: The authors suggested different mechanism between AMD3100 and JM#170. Since AMD3100 can block proliferation, as the authors claimed, AMD3100 should at least show some growth inhibition against this cell line. As AMD3100 was used as positive control but did not show efficacy as expected, the authors should prove anti-proliferation nature of AMD3100 to support this assumption.

5. Page 5, line 157: The authors have cited references to prove the relationship of CXCR4 in Bcr-Abl transformed pre-B cells. But it’s unknown if this triple knock out will bring other new genotype. Please include evidence to prove Bcr-Abl transformation can drive cell growth of this TKO-BM B cells.

6. Please provide more evidences for on-target validation of JM#170.

Reviewer 2 Report

Comments and Suggestions for Authors

In the present study, the author discussed the effect of CXCR4 inhibitors in BCR-ABL1+ ALL. The CXCR4 is essential for the survival of BCR-ABL1+ cells. They screened CXCR4 inhibitors including JM#170, which is specific for BCR-ABL1+ ALL and confirmed that JM#170 could emerge as a potent novel drug against Ph+ ALL. This target and inhibitor could be helpful for the clinicians to overcome the drug resistance in Ph+ ALL. However, there are some minor issues that need to be address:

1.     In introduction, it would be better to describe the disadvantages of current treatment of the Ph+ ALL and, why there is a need to develop a new inhibitor for Ph+ ALL? What is the rationale behind this study?

2.     Figures legends and descriptions of figures are confusing, need to be clearly explained.

3.     The author should have validated this finding further in primary cells to show JM#170 could emerge as a potent novel drug against Ph+ ALL.

4.     There are some typing errors and few incomplete sentences that need to be address carefully.

5.     In figure 5, include the combination index value to show the synergistic effect of combining imatinib with JM#170

Comments on the Quality of English Language

English need to improved 

Round 2

Reviewer 1 Report

Comments and Suggestions for Authors

The authors have well addressed on all the raised issue.

One minor suggestion is for the legend of figure 1. I understand the authors have well proved target validation of CXCR4. But all the figures and graphs in figure 1 are only related to cellular growth but not CXCR4 inhibition. The authors are suggested to rephrase it.

Author Response

Comments 1. The authors have well addressed on all the raised issue.

Answer 1. Thank you very much. We are happy that could address all the points properly.

Comments 2. One minor suggestion is for the legend of figure 1. I understand the authors have well proved target validation of CXCR4. But all the figures and graphs in figure 1 are only related to cellular growth but not CXCR4 inhibition. The authors are suggested to rephrase it.

Answer 1. Thank you for pointing this out. We have now removed the phrase "CXCR4 inhibition" from the legend of Figure 1 as well as the Supplementary Figure 1.